# Menopausal symptoms, physical activity level and quality of life of women living in the Mediterranean region

**Aya El Hajj[1], Nina Wardy[1◉], Sahar Haidar[1◉], Dana Bourgi[1‡], Mounia El Haddad[1‡], Daisy El Chammas[1‡], Nada El Osta[2,3,4], Lydia Rabbaa Khabbaz[5], Tatiana Papazian[1,5]***

**1** Department of Nutrition, Faculty of Pharmacy, Saint-Joseph University of Beirut, Beirut, Lebanon, **2** Department of Prosthodontics, Faculty of Dental Medicine, Saint Joseph University of Beirut, Beirut, Lebanon, **3** Laboratoire de Recherche Crâniofaciale, Unité de Santé Orale, Facuty of Dental Medecine, Saint Joseph University of Beirut, Beirut, Lebanon, **4** Centre de Recherche en Odontologie Clinique, University of Clermont Auvergne, EA, Clermont-Ferrand, France, **5** Laboratoire de pharmacologie, pharmacie clinique et contrôle de qualité des médicaments, Faculty of Pharmacy, Saint-Joseph University of Beirut, Beirut, Lebanon

◉ These authors contributed equally to this work.
‡ These authors also contributed equally to this work.
* tatiana.wahanian@usj.edu.lb

**Data Availability Statement:** All relevant data are within the manuscript and its Supporting Information files.

## Abstract

### Introduction

The main purpose of this study is to investigate the relationship between menopause related discomforts and the quality of life of Lebanese women correlated with the physical activity level, anthropometric, medical, sociodemographic and lifestyle variables, during mid-life.

### Materials and methods

This cross-sectional study was conducted among 1113 women, recruited from different Lebanese regions. The Menopause-Specific Quality of Life Questionnaire (MENQOL) was used to assess menopause related symptoms. Menopausal status was classified according to the World Health Organization's definition of menopause. The International Physical Activity Questionnaire was used to evaluate the physical activity level of participants. Anthropometric measurements were taken by the research team.

### Results

Participants were 49.53 ± 5.74 years old and had a mean body mass index of 26.69 ± 5.37 Kg/m². The highest mean scores of MENQOL were found in the physical and psychosocial domains (p<0.001). Peri-menopausal women had the highest mean scores in the vasomotor, physical and psychosocial domains (p<0.001), while postmenopausal and menopausal women in the sexual domain (p<0.001). Almost half the participants (45.4%) had low physical activity level, which was significantly and inversely correlated with vasomotor, psychosocial, physical and sexual MENQOL subdomains (p<0.001). Menopausal status, educational level, crowding and body mass indexes, marital status, smoking and alcohol intake were

**Funding:** The author(s) received no specific funding for this work.

**Competing interests:** The authors have declared that no competing interests exist.

among the factors that were significantly associated with the frequency and the severity of menopause related symptoms.

## Conclusion

Physical activity may play a protective role in attenuating climacteric symptoms and hence improve the quality of life of women during mid-life.

## Introduction

Female hormones play a central role in women's life. Their rise trigger puberty, allow them to experience the joy of motherhood and insure cardioprotective functions and bone health [1,2] However, after mid-forties, almost all women, irrespective of their cultural background and health conditions, begin to experience physical, psychological and emotional disturbances [3]. Those turmoils coincide with a progressive decline of female hormones, estrogen and progesterone, culminating to a total shutdown from the ovaries, diagnosed as menopause [3]. During this period, women present difficulties in accurately describing physical, psychosocial or sexual disturbances and report mainly hot flashes, nervousness, depression, insomnia, and general fatigue [3]. These vast arrays of symptoms progressively worsen the well-being of women, and affect, consequently, their quality of life (QOL) on a daily basis. Since the definition presented by the World Health Organization (WHO), in 1947, QOL refers to the "state of complete physical, mental and social well-being, and not merely the absence of disease and infirmity" of individuals, irrespective of their age, race and socio-economic status [4].

Nowadays, studying the correlations between psychosomatic symptoms, feelings, and the absolute well-being of women has been the center of research in a wide range of disciplines, going from medical to social sciences, based on specific scored tools. The Menopause-Specific Quality of Life Questionnaire (MENQOL), developed initially by Hilditch et al. in 1996, focuses on the QOL of women in midlife, during the past month. It has been validated and translated into many languages in various epidemiological and clinical settings. Four domains covering vasomotor, psychosocial, physical and sexual aspects are explored in this self-administered survey; they measure both the degree and the severity of several menopausal symptoms, from which a woman is affected by [5].

Lebanon is a small country located on the Eastern shore of the Mediterranean, known as the Middle-East. This region was frequently hit by civil wars and adopts Arabic as the official native language. However, the Lebanese societal and cultural habits are quite different from the Arabic neighboring nations, since Lebanese women are more socially empowered, which helps them play an important role in our modern society.

Despite the interest of researchers worldwide about the extent and type of symptoms experienced by women around menopause, only one single study published in 1999 has assessed this issue in our country, among 298 women, referring to a checklist of symptoms related to menopause [6].

However, living trends in our Lebanese society underwent a major lift nowadays, and those transitions positively encouraged women to become more active members. In addition, according to the latest statistics of the WHO published in 2016, life expectancy among Lebanese women reaches the age of seventy-eight, suggesting that they will probably live one quarter of their life in menopausal status [7].

In another hand, physical activity (PA) has been shown to enhance the QOL among menopausal women, probably because of its action on neuroendocrine balance and the release of endogenous opioids, which leads to decreased vasomotor symptoms [8,9]. Moreover, it can enhance psychosomatic well-being by improving the self esteem and the quality of sleep and decreasing the musculoskeletal pain and the menopausal complaints [10,11].

Hence, the main objective of this research is to investigate the relationship between menopause-related disconforts and the QOL of Lebanese women correlated with PA level, anthropometric, medical, sociodemographic and lifestyle variables.

## Materials and methods

This cross-sectional observational research was conducted among Lebanese females, aged between 40 and 60 years old, recruited from various geographic districts (urban and rural) of Lebanon, on different sites such as medical clinics, work places, banks, schools and housewives living in households. The research team decided to exclude all participants who experienced an induced menopause, due to several medical conditions such as hysterectomies, ovariectomies, radiation or chemotherapy and to only target those entering menopause naturally. This selection was applied because hormone replacement therapy, after such surgeries or medical conditions, interferes with menopausal status and thus might influence the results of this study. Pregnant or breastfeeding women and those suffering from mental illnesses, cognitive impairments or physical handicap were not eligible for participation. Field work was conducted between January and April 2018 by trained dietitians, on all Lebanese territories. The study protocol was approved by the Institutional Review Board of Saint-Joseph University at Beirut Lebanon (USJ-2018 / FP 46). All subjects gave their written consent prior to their participation in this study. Overall, around 1528 questionnaire were distributed among the research team to approach eligible candidates. Some women were either reluctant to join, non-lebanese, born Lebanese but living abroad, or having difficulties in reading Arabic. Hence, statistical analysis were performed on the total number of complete collected material (N = 1113).

Among the various questionnaires wordwidely published on the QOL of women during menopause, and after a thorough literature review, the research team decided to use an internationally validated questionnaire, the MENQOL, in its Arabic version, for communication facilities. Before proceeding, permission was solicited from the research team that applied this self-administered tool in its Arabic version [12]. It consists of a total of 29 items, divided into four domains: vasomotor (items 1–3), psychosocial (items 4–10), physical (items 11–26) and sexual (items 27–29). Answers, provided in a Likert-scale format, were displayed as "no" or "yes", with the latter spread from zero to six, respectively indicating the presence of the symptom and its degree, from being not bothersome to extremely bothersome. Calculations of each domain are computed separately and then summed up to reach the final score of MENQOL, as elucidated by the main developer of this tool [5].

PA level of each participant was defined via the International Physical Activity Questionnaire (IPAQ), in its short form, where subjects mention how much exercise they did in a typical week (S1 File). This self-administered questionnaire, used worldwide, assesses the overall PA over the last seven days, to categorize it as low, moderate or high [13].

The study material was tested on a sample of 20 women in a pilot study, prior to its official launch. The first section of the questionnaire, concerning anthropometric measurements, sociodemographic and medical data was filled by the research team, during face-to-face interviews. The socioeconomic status were the crowding index, the educational level and the current work status. The crowding index refers to the ratio of the total number of co-residents per household with the exception of newborn infants divided by the total number of rooms,

excluding the kitchen and the bathrooms. An index less than one suggested a household with high economic resources, and vice versa. Both the educational level and the crowding index are essentially needed in epidemiological studies to define the socioeconomic status of a sample.

Menopausal status was defined in accordance to the WHO's classification. To elucidate this distribution, women with regular menstrual bleeding during the last year were classified as premenopause, those with irregular bleeding during the last 12 months as perimenopause, and those with amenorrhea during the last year as menopause. Finally, women were classified as post-menopaused, if they had no menstrual bleeding from 1 year and above. Body mass index (BMI) was calculated as the actual weight, in kilograms, divided by height, in meters squared, relying on the anthropometric inputs (height, weight) measured respectively by a stadiometer and a digital scale, by the research team, the day of the recruitment. It was then categorized according to the WHO cut-off points: underweight if less than 18.5, normal if between 18.5 and 24.9, overweight if between 25 and 29.9 and obese from 30 and above [14]. Waist circumference (WC) was obtained via a non-stretchable tape measure, at the narrowest point area of the waist, above the umbilicus, at mid-distance between the lowest rib and the iliac crest. Values above 80 cm were considered as a cut-off for abdominal obesity, as defined by the International Diabetus Federation ethnic specific values for Europids women [15]. To determine the waist to hip ratio (WHR), WC was divided by the hip circumference, measured at the widest location of the hips. According to the WHO reference values, results higher than 0.85 were considered as a benchmark for metabolic syndrome [16].

The last sections (MENQOL and IPAQ) were filled by the participant herself.

## Statistical analysis

Means and standard deviations were used to describe the continuous variables and percentages to describe categorical variables. Kolmogorov-Smirnov tests were attributed to assess the normality of distribution of continuous variables, Chi-square tests were executed for comparison of percentages, analyses of variance followed by Tukey post hoc tests were used for the comparison of continuous variables between several groups and, lastly, Student t tests and Mann-Whitney tests were applied for the comparison of continuous variables between two groups. Pearson and Spearman correlation coefficients were computed to study the relationship between continuous variables. Univariate analysis were followed by multivariate analysis (multiple regression analyses) to evaluate explanatory factors associated with each MENQOL subdomain as the dependant variable. The variable "waist circumference" was removed from the model because it was highly correlated with BMI and waist/hip ratio. The variable "age" was removed from the model because it was highly correlated with menopausal status. Also, the categorical variable "profession" was not included because it is highly correlated with the level of education. In multivariate models, marital status was categorized as married and unmarried. The significance threshold retained corresponds to a value of p less than 0.05. All statistical analysis were performed using the SPSS statistical software package version 23, with a p value less than 0.05 considered as significant. The sample size was calculated according to the formula of Tabachnick and Fidell (2001) [17] that takes into consideration the number of explanatory variables to be included in the model: $N = 50 + 8m$ (m is the number of explanatory variables); Given that $m = 14$, a minimum of 162 women should be recruited in this study.

## Results

### Sociodemographic characteristics of the population

The study population consisted of 1113 Lebanese women with a mean age of 49.5 years (± 5.7 SD). The majority (81.7%) were married and almost half of them (45%) were employed, had a

university degree (45.8%), and a mean crowding index of 0.9 ± 0.4. More details concerning their socio-demographic characteristics are shown in S1 Table.

## Anthropometric measurements, lifestyle, health and menopausal status of the participants

The mean BMI, WC, hip circumference and WHR of the participants were respectively 26.7 kg/m$^2$ (± 5. 4 SD), 90.3 cm (± 13.1 SD), 103.7 cm (± 13.9 SD) and 0.9 (± 0.1 SD). Menopausal status of each woman was identified at the time of the survey from her response to the question of "*Have you reached menopause for more or less than one year*?".

Mean ages of both the first menstrual period and menopause were respectively 12.6 (± 1.5 SD) and 47.9 (± 4.5 SD) years. Reported answers were that almost half of the participants (46.2%) were at pre-menopause, 40.7% at post-menopause and 13.2% scattered between peri-menopause and menopause. All results together with the mean scores of MENQOL's subdomains are summarized in Table 1.

The clinical data according to the menopausal status is summarized in the supplementary file (S2 Table).

In terms of severity of symptoms, the most critical vasomotor, psychosocial, physical, and sexual symptoms were: hot flashes (48.9%), anxiety and nervousness (68.9%), memory problems (52.9%), pain in muscles and joints (72.3%), feeling of tiredness or exhaustion (73%), lower back pain (65%), and modifications in sexual desire (43.4%). Detailed results are summarized in S3 Table.

As presented in Table 2, vasomotor, psychosocial and physical symptoms were more often experienced by peri and post-menopausal women (p<0.001), whereas post-menopausal and menopausal women suffered mostly from sexual disturbances.

Furthermore, significant associations were observed among other variables such as marital status, educational level, occupation, crowding index, BMI, smoking, and alcohol consumption and the MENQOL subdomains. Detailed results are shown in Table 3.

## Physical activity level and QOL of the participants

The majority of the participants had low (45.4%) or moderate (44%) PA level, with pre-menopausal and menopausal women being most active, compared to those already in menopause (Table 4).

As presented in Table 5, women with the lowest PA level had the highest scores of MENQOL subdomains and vice versa (p<0.001).These results determine the inverse correlation between PA level and the frequency of menopausal symptoms and the positive association between PA and the QOL of women in mid-life.

Multiple regression models have found that after controlling other variables, PA remains negatively correlated with vasomotor, psychosocial, physical and sexual symptoms (p<0.001). Detailed results are shown in Tables 6–9.

## Discussion

The objective of this cross-sectional study was to investigate the QOL of Lebanese women by using the MENQOL and to correlate the results with the PA level, anthropometric, sociodemographic, lifestyle and medical characteristics among a large sample of Lebanese women.

Medical instances are incapable to predict the exact timing of menopause, since this transition is a gradual process and is influenced by genetic, cultural and individual profiles. Several studies have specified a range of 45 to 55 years [6,18–20]. In our sample, the mean age of menopause of the participants was 47.9 ± 4.5, similar to the statistics published in Saudi Arabia

**Table 1. Anthropometric measurements, lifestyle, health, menopausal status and mean scores of MENQOL sub-domains of the study group.**

| Characteristics (N = 1113) | Mean | SD |
|---|---|---|
| BMI[‡] (Kg/m$^2$) | 26.7 | 5.4 |
| Waist circumference (cm) | 90.3 | 13.1 |
| Hip circumference (cm) | 103.7 | 13.9 |
| Waist to Hip ratio | 0.9 | 0.1 |
| Age at first menstrual period (years) | 12.6 | 1.5 |
| Age at menopause | 47.9 | 4.5 |
| Number of children | 2.7 | 1.3 |
| | **N** | **%** |
| **BMI[‡]** | | |
| Underweight: <18.5 | 12 | 1.1 |
| Normal: 18.5–24.9 | 453 | 40.7 |
| Overweight: 25–29.9 | 424 | 38.1 |
| Obesity I: 30–34.9 | 158 | 14.2 |
| Obesity II: 35–39.9 | 44 | 4 |
| Obesity III: > 40 | 21 | 1.9 |
| **Waist circumference[*]** | | |
| Normal < 80cm | 215 | 19.3 |
| High ≥ 80 cm | 850 | 76.4 |
| **Waist to Hip ratio[*]** | | |
| Normal <0.85 | 414 | 38.9 |
| High ≥ 0.85 | 650 | 61.1 |
| **Chronic diseases** | | |
| Presence | 280 | 25.2 |
| Absence | 833 | 74.8 |
| **Type of disease** | | |
| Diabetes | 66 | 5.9 |
| Hypertension | 155 | 13.9 |
| Cardiovascular diseases | 37 | 3.3 |
| **Smoking** | | |
| Yes | 566 | 50.9 |
| No | 547 | 49.1 |
| **Alcohol consumption** | | |
| Never | 784 | 70.4 |
| ≤ twice per week | 294 | 26.4 |
| >3 times per week | 35 | 3.1 |
| **Current increase of appetite** | | |
| Yes | 500 | 44.9 |
| No | 613 | 55.1 |
| **Self-evaluation of health status** | | |
| Excellent/Good | 684 | 61.5 |
| Normal | 367 | 33 |
| Bad/Very bad | 62 | 5.6 |
| **Regularity of the ovarian menstrual cycle** | | |
| Regular | 941 | 84.5 |
| Irregular | 172 | 15.5 |
| **Menopausal status[¶]** | | |

*(Continued)*

**Table 1.** (Continued)

| | | | |
|---|---|---|---|
| Pre-menopause | | 514 | 46.2 |
| Peri-menopause | | 83 | 7.5 |
| Menopause | | 63 | 5.7 |
| Post-menopause | | 453 | 40.7 |
| **MENQOL subdomains** | **N** | **Mean** | **SD** |
| Vasomotor | 1113 | 2.4 | 1.7 |
| Psychosocial | 1113 | 3.0 | 1.5 |
| Physical | 1113 | 3.2 | 1.3 |
| Sexual | 1086[*] | 2.6 | 2.0 |

[*] missing values

[‡] BMI: body mass index

[¶] Menopausal status was classified according to the WHO definition of menopause

(48.3 ± 3) in 2015 [21] and in Egypt (48.9 ± 4) in 2017 [22]. However, this value is slightly lower than the one specified by Bener et al. (2014) in the Gulf region [20], and by Jaber et al. (2017) among Jordanian women [23]. The direct cause of these discrepancies concerning the beginning of the menopausal process in women from different countries is hard to interpret. Methodological aspects combined with ethnic and genetic diversity of humans around the world, as well as some memory bias regarding the exact age of menopause, are most probably the main factors of this divergence.

Significant associations were observed between the educational level, the crowding index and the physical and psychosocial domains of the MENQOL, mainly observed in less educated and more economically deprived women as seen among Iranian women in 2011 [9]. On the

**Table 2. Mean scores of MENQOL subdomains according to menopausal periods.**

| | | N | Mean ± SD | P value |
|---|---|---|---|---|
| **Vasomotor subdomain** | Pre-menopause | 514 | 1.8 ± 1.3 [a] | <**0.001**[*] |
| | Peri-menopause | 83 | 3.2 ± 2.2 [b] | |
| | Menopause | 63 | 2.7 ± 1.7 [b] | |
| | Post-menopause | 453 | 2.9 ± 1.8 [b] | |
| **Psychosocial subdomain** | Pre-menopause | 514 | 2.9 ± 1.6 [a] | <**0.001**[*] |
| | Peri-menopause | 83 | 3.7 ± 1.7 [b] | |
| | Menopause | 63 | 2.7 ± 1.3 [a] | |
| | Post-menopause | 453 | 3.0 ± 1.5 [a] | |
| **Physical subdomain** | Pre-menopause | 514 | 3.0 ± 1.3 [a] | <**0.001**[*] |
| | Peri-menopause | 83 | 3.7 ± 1.4 [b] | |
| | Menopause | 63 | 3.1 ± 1.3 [a] | |
| | Post-menopause | 453 | 3.3 ± 1.3 [a] | |
| **Sexual subdomain** | Pre-menopause | 500 | 2.2 ± 1.7 [a] | <**0.001**[*] |
| | Peri-menopause | 83 | 2.8 ± 1.9 [b] | |
| | Menopause | 62 | 2.9 ± 2.0 [b] | |
| | Post-menopause | 441 | 2.9 ± 2.1 [b] | |

[*]Statistical analyses were done with ANOVA with a p value <0.05 considered as significant.

a, b: different letters indicate the presence of significant letters with Tukey post hoc tests; mean ±SD with letter "a" have the significant lowest values, and letters "b" indicate the highest values

**Table 3. Mean scores of MENQOL subdomains according to sociodemographic, anthropometric and lifestyle factors.**

| | Vasomotor (N = 1113) | Psychosocial (N = 1113) | Physical (N = 1113) | Sexual (N = 1086) |
|---|---|---|---|---|
| **Age** | | | | |
| 40–44 years | 1.9 ± 1.4 [a] | 3.1 ± 1.5 | 3.0 ± 1.3 | 2.1 ± 1.7 [a] |
| 45–49 years | 2.3 ± 1.6 [b] | 3.1 ± 1.7 | 3.2 ± 1.4 | 2.5 ± 1.9 [a, b] |
| 50–54 years | 2.8 ± 1.8 [c] | 3.0 ± 1.5 | 3.2 ± 1.3 | 2.6 ± 1.9 [b] |
| 55–59 years | 2.8 ± 1.8 [c] | 2.8 ± 1.4 | 3.2 ± 1.3 | 3.1 ± 2.2 [c] |
| P value | **<0.001**[*] | 0.336 | 0.231 | **<0.001**[*] |
| **Educational level** | | | | |
| Primary | 2.8 ± 1.7 | 3.6 ± 1.6 [b] | 3.7 ± 1.3 [c] | 3.1 ± 2.2 [b] |
| Complementary | 2.6 ± 1.9 | 3.4 ± 1.6 [b] | 3.6 ± 1.4 [b, c] | 2.7 ± 2.1 [a, b] |
| Secondary | 2.4 ± 1.7 | 3.2 ± 1.6 [b] | 3.3 ± 1.3 [b] | 2.9 ± 2.0 [b] |
| University | 2.3 ± 1.6 | 2.6 ± 1.4 [a] | 2.9 ± 1.2 [a] | 2.3 ± 1.7 [a] |
| P value | 0.101 | **<0.001**[*] | **<0.001**[*] | **<0.001**[*] |
| **Marital status** | | | | |
| Single | 2.3 ± 1.7 | 2.7 ± 1.6 [a] | 2.9 ± 1.3 | 1.6 ± 1.2 [a] |
| Married | 2.4 ± 1.7 | 3.0 ± 1.5 [a, b] | 3.2 ± 1.3 | 2.8 ± 2.0 [b] |
| Divorced | 2.9 ± 1.9 | 3.4 ± 1.8 [a, b] | 3.2 ± 1.3 | 2.1 ± 1.7 [a,b] |
| Widow | 2.5 ± 1.7 | 3.3 ± 1.5 [b] | 3.3 ± 1.2 | 1.7 ± 1.6 [a] |
| P value | 0.233 | 0.050 | 0.122 | **<0.001**[*] |
| **Crowding index** | | | | |
| >1 | 2.6 ± 1.8 | 3.3 ± 1.6 [b] | 3.5 ± 1.3 [b] | 2.9 ± 2.0 [b] |
| 1 | 2.5 ± 1.8 | 3.2 ± 1.6 [b] | 3.3 ± 1.3 [b] | 2.5 ± 2.0 [a,b] |
| <1 | 2.3 ± 1.6 | 2.8 ± 1.5 [a] | 3.0 ± 1.3 [a] | 2.5 ± 1.9 [a] |
| P value | 0.058 | **<0.001**[*] | **<0.001**[*] | **0.013**[*] |
| **Occupation** | | | | |
| Employed | 2.3 ± 1.6 [b] | 2.8 ± 1.4 [a] | 3.0 ± 1.3 [a] | 2.4 ± 1.8 [a] |
| Unemployed | 2.5 ± 1.8 [b] | 3.2 ± 1.6 [b] | 3.3 ± 1.3 [a, b] | 2.8 ± 2.0 [b] |
| Retired | 2.1 ± 1.7 [a] | 3.1 ± 1.6 [b] | 3.1 ± 1.2 [a] | 2.4 ± 2.0 [a] |
| P value | **0.045**[*] | **<0.001**[*] | **0.001**[*] | **<0.001**[*] |
| **BMI[¶]** | | | | |
| Underweight: <18.5 | 2.0 ± 1.2 | 3.2 ± 1.4 | 3.1 ± 1.36 | 3.3 ± 2.0 |
| Normal: 18.5–24.9 | 2.3 ± 1.6 | 2.8 ± 1.5 | 2.9 ± 1.24 | 2.4 ± 1.8 |
| Overweight: 25–29.9 | 2.5 ± 1.7 | 3.0 ± 1.5 | 3.2 ± 1.30 | 2.6 ± 1.9 |
| Obesity I: 30–34.9 | 2.8 ± 1.8 | 3.4 ± 1.6 | 3.8 ± 1.28 | 2.8 ± 2.2 |
| Obesity II: 35–39.9 | 2.7 ± 1.8 | 3.4 ± 1.7 | 3.8 ± 1.14 | 3.2 ± 2.2 |
| Obesity III: >40 | 2.5 ± 1.7 | 3.7 ± 1.8 | 4.1 ± 1.20 | 3.3 ± 2.1 |
| P value | **0.020**[*] | **<0.001**[*] | **<0.001**[*] | **0.010**[*] |
| **Waist circumference** | | | | |
| Normal <80cm | 2.3 ± 1.7 | 2.8 ± 1.4 | 2.9 ± 1.3 | 2.6 ± 1.9 |
| High ≥ 80 cm | 2.5 ± 1.7 | 3.1 ± 1.6 | 3.3 ± 1.3 | 2.6 ± 1.9 |
| P value | 0.051 | 0.050 | **<0.001**[*] | 0.697 |
| **Waist–hip ratio** | | | | |
| Normal <0.85 | 2.3 ± 1.7 | 3.0 ± 1.5 | 3.1 ± 1.3 | 2.7 ± 2.0 |
| Obese ≥ 0.85 | 2.5 ± 1.7 | 3.0 ± 1.6 | 3.2 ± 1.3 | 2.6 ± 1.9 |
| P value | **0.043**[*] | 0.971 | 0.120 | 0.711 |
| **Current increase of appetite** | | | | |
| Yes | 2.8 ± 1.8 | 3.2 ± 1.6 | 3.0 ± 1.3 | 2.8 ± 2.0 |
| No | 2.1 ± 1.6 | 2.8 ± 1.5 | 3.0 ± 1.3 | 2.4 ± 1.9 |

*(Continued)*

**Table 3.** (Continued)

| | Vasomotor (N = 1113) | Psychosocial (N = 1113) | Physical (N = 1113) | Sexual (N = 1086) |
|---|---|---|---|---|
| P value | **<0.001**[*] | **<0.001**[*] | **<0.001**[*] | **0.004**[*] |
| **Self-evaluation of health** | | | | |
| Excellent/Good | 2.1 ± 1.5 [a] | 2.6 ± 1.3 [a] | 2.8 ± 1.1 [a] | 2.3 ± 1.8 [a] |
| Normal | 2.9 ± 1.8 [b] | 3.5 ± 1.6 [b] | 3.7 ± 1.3 [b] | 2.9 ± 2.1 [b] |
| Bad/very bad | 3.2 ± 2.3 [c] | 4.5 ± 1.9 [c] | 4.6 ± 1.2 [c] | 3.9 ± 2.4 [c] |
| P value | **<0.001**[*] | **<0.001**[*] | **<0.001**[*] | **<0.001**[*] |
| **Smoking** | | | | |
| Yes | 2.6 ± 1.8 | 3.20 ± 1.55 | 3.27 ± 1.28 | 2.76 ± 2.02 |
| No | 2.3 ± 1.6 | 2.83 ± 1.51 | 3.10 ± 1.34 | 2.45 ± 1.87 |
| P value | **<0.001**[*] | **<0.001**[*] | **0.030**[*] | **0.008**[*] |
| **Alcohol consumption** | | | | |
| Never | 2.4 ± 1.7 [a] | 3.1 ± 1.6 [b] | 3.2 ± 1.3 [b] | 2.7 ± 2.0 [b] |
| ≤ twice/week | 2.6 ± 1.7 [b] | 2.9 ± 1.4 [a] | 3.1 ± 1.3 [b] | 2.3 ± 1.8 [a, b] |
| > 3 times/week | 2.7 ± 2.1 [b] | 2.7 ± 1.6 [a] | 2.9 ± 1.4 [a] | 2.2 ± 1.7 [a] |
| P value | **0.047**[*] | 0.084 | 0.054 | **0.004**[*] |

¶BMI: body mass index

[*]Statistical analyses were done with ANOVA, Student t tests/Mann-Whitney tests with a p value <0.05 considered as significant.

a, b, c: different letters indicate the presence of significant letters with Tukey post hoc tests; mean ±SD with letter "a" have the significant lowest values, letters "b" have the intermediate values and letters "c" the highest values.

other hand, a better QOL was achieved among educated and those belonging to higher socio-economic level. The main reason behind those differences is that educated women are continuously more concerned about their actual condition and eager to find solutions either thru personal research or with the help of professionals, and often have better access to health care plans. Almost half the women of our samples had a university degree and were employed (45%). Those two factors contribute greatly to higher feelings of self-esteem, independence and self confidence, which can explain the better QOL on the psychosocial level [24,25].

Anthropometric measurements showed that the mean BMI, WC, and WHR were exceeding the normal values recommended for this subgroup of population. These results concord with those published in Lebanon in two distinct studies in 2003 and 2016, and highlight a major

**Table 4. Physical activity level of the participants and their classification according to their menopausal status.**

| | N | | | | % |
|---|---|---|---|---|---|
| **Physical activity level** | | | | | |
| **Low** | 505 | | | | 45.4 |
| **Moderate** | 490 | | | | 44 |
| **High** | 118 | | | | 10.6 |
| | **Menopausal status** | | | | |
| | Pre-menopause N = 514 | Peri-menopause N = 83 | Menopause N = 63 | Post-menopause N = 453 | P value |
| **Physical activity level** | | | | | **0.003**[*] |
| **Low** | 210 (40.9%) | 34 (41.0%) | 25 (39.7%) | **236 (52.1%)** | |
| **Moderate** | 239 (46.5%) | 45 (54.2%) | 29 (46.0%) | 177 (39.1%) | |
| **High** | **65 (12.6%)** | 4 (4.8%) | **9 (14.3%)** | 40 (8.8%) | |

[*]Statistical analyses were done with Chi-square test, with a p value <0.05 considered as significant

**Table 5. Mean scores of MENQOL subdomains according to physical activity level of participants.**

| | Vasomotor subdomain (N = 1113) | Psychosocial subdomain (N = 1113) | Physical subdomain (N = 1113) | Sexual subdomain (N = 1086) |
|---|---|---|---|---|
| **Physical activity level** | | | | |
| Low | 2.79 ± 1.79 [c] | 3.47 ± 1.64 [c] | 3.61 ± 1.35 [c] | 3.04 ± 2.09 [c] |
| Moderate | 2.31 ± 1.65 [b] | 2.83 ± 1.39 [b] | 3.08 ± 1.14 [b] | 2.39 ± 1.83 [b] |
| High | 1.56 ± 1.03 [a] | 1.84 ± .86 [a] | 1.87 ± 0.65 [a] | 1.62 ± 1.17 [a] |
| P value | <**0.001**[*] | <**0.001**[*] | <**0.001**[*] | <**0.001**[*] |

a, b, c: different letters indicate the presence of significant letters with Tukey post hoc tests; mean ±SD with letter "a" have the significant lowest values, letters "b" have the intermediate values and letters "c" have the highest values.

[*] Statistical analyses were done with ANOVA, with a p value <0.05 considered as significant.

future public health issue, associated with excess body weight and abdominal obesity in this population's category [26,27]. A high BMI value was significantly associated with poor QOL in all domains, compatible with the results of other studies published elsewhere [28–30]. Excess weight can be the major cause of physical distress and is manifested mostly by back pain and psychological worries, consequently influencing the QOL of women.

Surprisingly, in our study, alcohol consumption improved the QOL of women in the sexual aspect, in contrast to non-drinkers. This can be explained by the mood boosting effect of alcohol on dopamine secretions, leading to a total body relaxation. However, alcohol intake can also aggravate vasomotor symptoms like hot flashes and sweats, as proven in our study [31–33].

Overall, the frequency of symptoms invalidating the QOL of participants concerned primarily both the physical (fatigue, muscular and joint pain) and the psychosocial (anxiety, nervousness and memory loss) domains. Avoidance of sexual intimacy and modification in sexual desire were frequently encountered in our sample, reaching respectively 37% and 43% of participants. On the other hand, vasomotor symptoms such as hot flashes and sweats were reported less bothersome by Lebanese women, compared to those of Jahanfar et al. (2006) among Malaysian women, who suffer mostly from musculoskeletal pain (84.3%), anxiety (71.4%), physical and mental discomfort (67.2%), hot flashes and sweats (67.1%) [34]. Similar results were observed among women in Srilanka [35], whereas a predominance of

**Table 6. Multiple regression models of factors associated with the vasomotor subdomain.**

| | Non standardized coefficient | | Standardized coefficient | t | P value | Partial correlation |
|---|---|---|---|---|---|---|
| | **B** | **Standard error** | **Beta** | | | |
| Crowding index | 0.221 | 0.141 | 0.051 | 1.562 | 0.119 | 0.051 |
| Education level | 0.025 | 0.058 | 0.014 | 0.426 | 0.670 | 0.014 |
| BMI [‡] | 0.003 | 0.010 | 0.010 | 0.313 | 0.755 | 0.010 |
| Waist–Hip ratio | -0.022 | 0.562 | -0.001 | -0.040 | 0.968 | -0.001 |
| Smoking | 0.216 | 0.107 | 0.063 | 2.011 | **0.045**[*] | 0.065 |
| Alcohol consumption | -0.346 | 0.099 | -0.105 | -3.502 | <**0.001**[*] | -0.113 |
| Menopausal status | 0.215 | 0.028 | 0.233 | 7.698 | <**0.001**[*] | 0.243 |
| Number of children | 0.054 | 0.041 | 0.039 | 1.291 | 0.197 | 0.042 |
| Current increase of appetite | 0.482 | 0.104 | -0.139 | 4.642 | <**0.001**[*] | 0.150 |
| Auto-evaluation of health status | 0.270 | 0.050 | 0.168 | 5.443 | <**0.001**[*] | 0.175 |
| **Physical activity level** | **-0.377** | **0.082** | **-0.135** | **-4.386** | <**0.001**[*] | **-0.141** |

[‡]BMI: body mass index

**Table 7. Multiple regression of factors associated with the psychosocial subdomain.**

| | Non standardized coefficient | | Standardized coefficient | t | P value | Partial correlation |
|---|---|---|---|---|---|---|
| | B | Standard error | Beta | | | |
| Education level | -0.220 | 0.052 | -0.135 | -4.200 | <**0.001**[*] | -0.133 |
| Marital status | -0.204 | 0.154 | -0.039 | -1.321 | 0.187 | -0.042 |
| Crowding index | 0.257 | 0.123 | 0.066 | 2.095 | **0.036**[*] | 0.067 |
| BMI (Kg/m$^2$) [‡] | 0.009 | 0.009 | 0.032 | 1.084 | 0.279 | 0.035 |
| Smoking | 0.122 | 0.093 | 0.040 | 1.315 | 0.189 | 0.042 |
| Alcohol consumption | -0.043 | 0.085 | -0.015 | -0.505 | 0.614 | -0.016 |
| Menopausal status | -0.079 | 0.024 | -0.096 | -3.239 | **0.001**[*] | -0.103 |
| Number of children | -0.044 | 0.038 | -0.036 | -1.167 | 0.244 | -0.037 |
| Current increase of appetite | 0.214 | 0.090 | 0.069 | 2.381 | **0.017**[*] | 0.076 |
| Auto-evaluation of health status | 0.354 | 0.043 | 0.247 | 8.269 | <**0.001**[*] | 0.255 |
| **Physical activity level** | **-0.526** | **0.072** | **-0.221** | **-7.469** | <**0.001**[*] | **-0.232** |

[‡]BMI: body mass index

musculoskeletal pain was observed in Turkish women [28]. On the other hand, women in the Gulf region and Qatar suffered mostly from symptoms in the physical and psychosocial domains [20]. Our results join those published among Egyptian women who suffer mostly from joint and muscular discomfort (82.1%), physical and mental exhaustion (69.6%) and hot flashes (53.6%) [22]. In contrast, studies conducted in Saudi Arabia, Jordan and Iran confirmed that women endure more in the sexual domain rather than the psychosocial domain [21,23,9]. These differences in the frequency and the severity of symptoms are attributed to variances in sociocultural, ethnic, genetic and environmental factors, influencing the way a woman cope with this changeover and how much her spouse and family circle are supportive.

Peri-menopausal women in our sample suffered mostly from climacteric symptoms, whereas those already in menopause and post-menopause reported failure in their sexual life (p <0.001). Our findings join a study published in 2014, using the MENQOL, in Qatari women [36]. This issue can be attenuated by seeking medical care and by being more communicative and open within their couple.

**Table 8. Multiple regression of factors associated with the physical subdomain.**

| | Non standardized coefficient | | Standardized coefficient | t | P value | Partial correlation |
|---|---|---|---|---|---|---|
| | B | Standard error | Beta | | | |
| Education level | -0.139 | 0.042 | -0.100 | -3.278 | **0.001**[*] | -0.106 |
| Marital status | 0.121 | 0.127 | 0.026 | 0.955 | 0.340 | 0.031 |
| Crowding index | 0.250 | 0.099 | 0.076 | 2.517 | **0.012**[*] | 0.082 |
| BMI (kg/m$^2$) [‡] | 0.030 | 0.007 | 0.120 | 4.255 | <**0.001**[*] | 0.138 |
| Waist-Hip ratio | 0.481 | 0.394 | 0.033 | 1.220 | 0.223 | 0.040 |
| Smoking | 0.059 | 0.076 | 0.022 | 0.776 | 0.438 | 0.025 |
| Alcohol consumption | 0.025 | 0.069 | 0.010 | 0.364 | 0.716 | 0.012 |
| Menopausal status | -0.009 | 0.020 | -0.012 | -0.444 | 0.657 | -0.014 |
| Number of children | -0.005 | 0.031 | -0.005 | -0.171 | 0.864 | -0.006 |
| Current increase of appetite | 0.248 | 0.073 | 0.094 | 3.386 | **0.001**[*] | 0.110 |
| Auto-evaluation of health status | 0.385 | 0.035 | 0.314 | 10.979 | <**0.001**[*] | 0.337 |
| **Physical activity level** | **-0.499** | **0.057** | **-0.249** | **-8.721** | <**0.001**[*] | **-0.274** |

[‡]BMI: body mass index

**Table 9. Multiple regression of factors associated with the sexual subdomain.**

| | Non standardized coefficient | | Standardized coefficient | t | P value | Partial correlation |
|---|---|---|---|---|---|---|
| | B | Standard error | Beta | | | |
| Education level | -0.078 | 0.070 | -0.037 | -1.115 | 0.265 | -0.036 |
| Marital status | 1.210 | 0.217 | 0.170 | 5.580 | <**0.001**[*] | 0.176 |
| Crowding index | 0.144 | 0.166 | 0.029 | 0.870 | 0.385 | 0.028 |
| BMI (kg/m$^2$) [‡] | 0.014 | 0.012 | 0.038 | 1.211 | 0.226 | 0.039 |
| Smoking | 0.208 | 0.123 | 0.053 | 1.693 | 0.091 | 0.054 |
| Alcohol consumption | -0.311 | 0.114 | -0.083 | -2.732 | **0.006**[*] | -0.087 |
| Menopausal status | 0.183 | 0.033 | 0.172 | 5.577 | <**0.001**[*] | 0.176 |
| Number of children | -0.072 | 0.051 | -0.046 | -1.403 | 0.161 | -0.045 |
| Current increase of appetite | 0.200 | 0.121 | 0.050 | 1.659 | 0.097 | 0.053 |
| Auto-evaluation of health status | 0.222 | 0.058 | 0.120 | 3.827 | <**0.001**[*] | 0.122 |
| **Physical activity level** | **-0.484** | **0.095** | **-0.160** | **-5.091** | <**0.001**[*] | **-0.161** |

[‡]BMI: body mass index

According to the majority of studies conducted worldwide, peri-menopause is the worst period affecting negatively the QOL of women. The reason behind is the instability of female hormones, especially estrogen, that exacerbates vasomotor manifestations [37]. Besides, peri-menopausal women are psychologically more vulnerable to face this transition in their life, and are aware of the physical outcomes such as weight gain and changes in skin quality associated with menopause. Hence, the importance of health strategies by multi-disciplinary professionals (dietitians, midwives, nurses, psychologists and doctors) to aid women to find solutions attenuating the negative repercussions of menopause on their everyday life [38].

After adjusting for cofounding variables, statistically significant correlations exist between PA level, menopausal symptoms and the QOL of women during midlife. Women who were physically more active, were suffering less from vasomotor, psychosocial, physical and sexual symptoms compared to sedentary ones. These findings are consistent with those of Skrzypulec et al. (2010) [39], Canário et al. (2012) [40] (11) and Dabrowsaka-Ga las et al. (2019) [41] who showed that PA correlates inversely with menopausal symptoms. They are also compatible with the results of Elvasky et al. (2005) (8) and Mansikkamaki et al. (2015) [42] who reported that PA improves the QOL of women during midlife. In fact, the mechanisms underlying the effects of PA on women's QOL during midlife are still uncertain. PA can decrease vasomotor symptoms because of its effect on endocrine balance in the autonomic system and on the release of endogenous opioids and may increase the production of hypothalamic and peripheral ß-endorphins. Therefore, it helps stabilize body temperature, heart and respiratory rate, ameliorate the sensitivity to pain, and diminish the risk of hot flashes [39]. In addition, regular PA is associated with improved self-esteem and better quality of sleep, decreased menopausal complaints and subsequent improvement in psychosomatic well-being [8,40]. This association between PA and psychosomatic symptoms can be mediated by several psychological and physiological mechanisms, including the diversion of stressful stimuli, increased levels of endorphins and improvement in self-efficacy and cerebral aminergic synaptic transmission [43–45]. Moreover, PA can improve women's sexual life by reducing the vaginal dryness and increasing the lubrication, due to a better vascularization and peripheral oxygenation [41,46]. Concerning the musculoskeletal health, it reduces muscle loss, increases bone density, maintains motor skills and decreases risk of fractures [47]. Finally, PA helps to stabilize and prevent the increase of body weight and WC [48]. Hence, all these results stress on the benefits of PA among menopausal women.

Some limitations concerning this research merit our attention. The design of this study was cross-sectional, aiming to evaluate the QOL of women before, during and after menopause, associated with personal variables. This type of epidemiological study does not allow to establish the impact of those factors over time. This issue can be corrected in conducting a longitudinal study in the near future. Another limitation concerns the use of a self-administered instrument, the IPAQ, to evaluate the PA level of participants. Research in this aspect favors the use of accelerometers to define more accurately the PA status compared to subjective tools [49]. However, the reliability and the validity of the IPAQ and the use of accelerometers were not in the scope of this project. Although Lebanese women enjoy more freedom and are more assertive compared to women in neighboring countries, some participants were reluctant to answer the questions related to sex in the MENQOL, since this issue was considered by them as a taboo and highly personal. The research team followed strict instructions not to interfere during self-reporting and insured women on several occasion about the total confidentiality and anonymity of the results. Hence, women were less intimidated and more likely to respond. This emerged as an added-value in the response rate of this section. Our sampling strategy was not representative of all Lebanese women, however unlike the previous one conducted in 1999 among 298 Lebanese women [6], its major strength is that it englobed a large number of participants (N = 1113) from different educational, marital and religious backgrounds and of various stages of the menopausal transition. Data emerged from this study can help researches establish future perspectives on national level by applying proper randomization and stratification and adding more parameters, such as blood tests to check levels of hormones. Another strength are the anthropometric measurements taken directly by the research team with standardized techniques, limiting the errors associated with self-reporting or under-reporting. Finally, the use of a scored tool (MENQOL) permitted to evaluate different parameters in order to assess the capacity of a woman to deal with this unique experience.

## Conclusion

Women climbing into menopause suffer uncommon symptoms, disrupting their well-being directly and their family circle indirectly with negative repercussions on their productivity and health during adulthood, deteriorating their QOL. The majority of public health campaigns conducted in our country focus more on maternal and neonatal health. The findings of our research show that exercise is effective in reliving menopausal symptoms. Thus, this study highlights the importance of health-promoting strategies conducted by health professionals such as nurses, dietitians, mid-wives and doctors, by counseling women at mid-life to adopt healthier and active lifestyles, coupled with relaxation practices to remedy those symptoms and improve their QOL.

## Supporting information

**S1 File. The International Physical Activity Questionnaire (IPAQ) in Arabic.**
(DOCX)

**S1 Table. Sociodemographic characteristics of the study group (N = 1113).**
(DOCX)

**S2 Table. Clinical data according to the menopausal status of the participants (N = 1113).**
(DOCX)

**S3 Table. The distribution of the menopausal symptoms according to the menopausal status (N = 1113).**
(DOCX)

## Acknowledgments

The authors gratefully acknowledge the participants.

## Author Contributions

**Conceptualization:** Tatiana Papazian.

**Data curation:** Aya El Hajj, Nina Wardy, Sahar Haidar, Dana Bourgi, Nada El Osta, Tatiana Papazian.

**Formal analysis:** Nada El Osta.

**Investigation:** Aya El Hajj, Sahar Haidar, Mounia El Haddad, Daisy El Chammas, Tatiana Papazian.

**Methodology:** Aya El Hajj, Nina Wardy, Sahar Haidar, Tatiana Papazian.

**Project administration:** Tatiana Papazian.

**Software:** Nada El Osta.

**Supervision:** Lydia Rabbaa Khabbaz, Tatiana Papazian.

**Validation:** Nada El Osta, Lydia Rabbaa Khabbaz, Tatiana Papazian.

**Writing – original draft:** Aya El Hajj.

**Writing – review & editing:** Tatiana Papazian.

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
