## [Decision Letter · Decision Letter 0]

30 Oct 2019

PONE-D-19-25613

Menopausal symptoms, physical activity level and quality of life of women living in the Mediterranean region.

PLOS ONE

Dear Dr Papazian,

Thank you for submitting your manuscript to PLOS ONE. After careful consideration, we feel that it has merit but does not fully meet PLOS ONE’s publication criteria as it currently stands. Therefore, we invite you to submit a revised version of the manuscript that addresses the points raised during the review process.

We would appreciate receiving your revised manuscript by Dec 14 2019 11:59PM. To enhance the reproducibility of your results, we recommend that if applicable you deposit your laboratory protocols in protocols.io, where a protocol can be assigned its own identifier (DOI) such that it can be cited independently in the future. For instructions see: http://journals.plos.org/plosone/s/submission-guidelines#loc-laboratory-protocols

We look forward to receiving your revised manuscript.

Kind regards,

Anderson Saranz Zago, PhD

Academic Editor

PLOS ONE

Journal Requirements:

3. Please clarify in your Methods section whether the questionnaire is published under a CC-BY license, or whether you obtained permission from the publisher to reproduce the questionnaire in this manuscript. Please explain any copyright or restrictions on this questionnaire

4. Please change your reference to "p=0.000" to "p<0.001" or as similarly appropriate, as p values cannot equal zero.

5. Please amend your manuscript to include your abstract after the title page.

6. Please ensure that you include a title page within your main document. You should list all authors and all affiliations as per our author instructions and clearly indicate the corresponding author.

Additional Editor Comments:

Dear Authors

The reviewers of the manuscript pointed many aspects that must be revised.

If you still have interest in public your manuscript in a PLOS ONE journal, please submit again with changes and comments.

Best,

Anderson Zago

Academic Editor

Reviewers' comments:

Reviewer's Responses to Questions

**Comments to the Author**

1. Is the manuscript technically sound, and do the data support the conclusions?

Reviewer #1: Yes

Reviewer #2: Yes

2. Has the statistical analysis been performed appropriately and rigorously? 

Reviewer #1: Yes

Reviewer #2: Yes

3. Have the authors made all data underlying the findings in their manuscript fully available?

Reviewer #1: Yes

Reviewer #2: Yes

4. Is the manuscript presented in an intelligible fashion and written in standard English?

Reviewer #1: Yes

Reviewer #2: Yes

5. Review Comments to the Author

Reviewer #1: I read with great interest the study "Menopausal symptoms, physical activity level and quality of life of women living in the

Mediterranean region" by Papazian et al.

The paper is well written, and the study deals with an important topic (i.e., menopause and quality of life), which has been extensively studied in western populations, but not too much in the Middle-East. Since there might be cultural- and social-related differences the way women cope with menopause, I think the current study is original in the sense that it originally provides a large amount of updated data related to quality of life, physical activity, anthropometrics in a rather understudied population.

The authors should be praised for their hard work and I have only minor questions/suggestions for improvements.

1) In introduction I would change "often failed to fully describe" to " a less strong sentence (maybe "present difficulties in accurately describing").

2) It is not clear if "hormone replacement therapy" was/wasn't a exclusion criteria.

3) I feel sorry for the 415 questionnaires that had to be removed from the final pool due to incomplete data. However, I wonder if the researchers couldn't at least compute the QOL domains that were filled by the participants and not exclude the whole data.

4) I also wonder if excluding the women that did not answer sexual-related questions couldn't be faced as a bias of the study. Couldn't part of the results of these women (the one that they answered) be included in the final analysis?

5) Overall, I don't think it is necessary to explain in details all the questionnaires (e.g., lines 70-77 for MENQOL; lines 80-81 for IPAQ, and so on). Authors should keep it simple and provide references for further details.

6) Tables 1 and 2 could be merged.

7) On all Tables authors should make it very clear the meaning/role of the overwritten letters.

8) Discussion on lines 232-243 is not necessary as it does not related to the objective of the study.

9) I think the physical activity is a nice outcome of the study, and this was independently associated with most of the subdomains of quality of life in the participants, which reinforce the importance of this variable to the study conclusions. However, authors should acknowledge the limitations of using a questionnaire to assess physical activity as it might present inaccuracies in comparison with objective measurements (e.g., accelerometers).

10) Another limitation is the absence of blood tests to check levels of FSH and estrogen hormones (even if it was done in a sub sample). I would include this in the limitation section but I don't think this limitation reduce the quality of the study.

11) It is not clear if there was a sampling strategy (and sample calculation) to generate a sample representative of all Lebanon areas. If not, I would include this as a limitation of the study.

Reviewer #2: I appreciate the effort of the authors to conduct the study for reporting the relationship between physical activity level, menopausal symptoms and the quality of life of Lebanese women. After completling my review, several concerns should be revised to fulfill the quality of the study.

General Comments

Please standardize the objective of the study on the different topics of this article (i.e. abstract, introduction and, discussion).

Please review your reference as more than half of the references listed are very outdated. If able, prefer to keep references not older than 10 years.

In addition, some English mistakes need to be corrected through all the manuscript.

Specific Comments

Abstract

In methods, please provide information regarding the classification of menopausal status.

Introduction

Please add bibliographic references in the first sentences of the first paragraph.

Please add the meaning of “MENQQOL”.

Methods

Please provide information regarding the randomization process of the different regions recruited.

Was any participant using hormone replacement therapy?

The authors do not mention the application of a cognitive test. Participants with cognitive impairment may have difficulty with the questionnaires.

Results

Please review this session. Some table results are not consistent with the text.

Please review the table captions. It is not possible to identify statistical differences among/between groups.

Please add to table 4 the results of the physical activity level data analysis.

Discussion

1° Paragraph: Please discuss more about the impact of anthropometric measurements (e.g. BMI, WC and WHR) on participants of this current study. Or join this part with the beginning of the 4° paragraph.

Please discuss more about the relationship between the results of the activity physical level and the different subdomains of women's quality of life.

Please discuss whether the results reach the level of clinical significance.

6. PLOS authors have the option to publish the peer review history of their article (what does this mean?). If published, this will include your full peer review and any attached files.

Reviewer #1: Yes: Tiago Peçanha

Reviewer #2: No

---

## [Author Response · Author response to Decision Letter 0]

22 Jan 2020

Dear editor and reviewers,

On behalf of all the authors, we highly appreciate your valuable comments and recommendations, that were fully taken into consideration in the final revised submitted manuscript. Hope it will meet your requirements. 

Journal Requirements:

1. When submitting your revision, we need you to address these additional requirements. Please ensure that your manuscript meets PLOS ONE's style requirements, including those for file naming. The PLOS ONE style templates can be found at http://www.journals.plos.org/plosone/s/file?id=wjVg/PLOSOne_formatting_sample_main_body.pdf andhttp://www.journals.plos.org/plosone/s/file?id=ba62/PLOSOne_formatting_sample_title_authors_affiliations.pdf

Done.

Done. Two supplementary tables were added and mentioned in the text.

3. Please clarify in your Methods section whether the questionnaire is published under a CC-BY license, or whether you obtained permission from the publisher to reproduce the questionnaire in this manuscript. Please explain any copyright or restrictions on this questionnaire

No copyright or restrictions were imposed. 

After conducting the literature review and before proceeding, permission was solicited from the research team that applied this self-administered tool in its Arabic version. 

4. Please change your reference to "p=0.000" to "p<0.001" or as similarly appropriate, as p values cannot equal zero.

Done. All p values were corrected from p=0.000 to p<0.001.

5. Please amend your manuscript to include your abstract after the title page.

Done.

6. Please ensure that you include a title page within your main document. You should list all authors and all affiliations as per our author instructions and clearly indicate the corresponding author.

Done. 

Review Comments to the Author

Reviewer #1: I read with great interest the study "Menopausal symptoms, physical activity level and quality of life of women living in the

Mediterranean region" by Papazian et al.

The paper is well written, and the study deals with an important topic (i.e., menopause and quality of life), which has been extensively studied in western populations, but not too much in the Middle-East. Since there might be cultural- and social-related differences the way women cope with menopause, I think the current study is original in the sense that it originally provides a large amount of updated data related to quality of life, physical activity, anthropometrics in a rather understudied population.

The authors should be praised for their hard work and I have only minor questions/suggestions for improvements.

1) In introduction I would change "often failed to fully describe" to " a less strong sentence (maybe "present difficulties in accurately describing").

Done.

2) It is not clear if "hormone replacement therapy" was/wasn't a exclusion criteria.

Hormone replacement therapy was an exclusion criteria.

3) I feel sorry for the 415 questionnaires that had to be removed from the final pool due to incomplete data. However, I wonder if the researchers couldn't at least compute the QOL domains that were filled by the participants and not exclude the whole data.

Unfortunately , the removal of some of the questionnaires from the final pool was not well elucidated in our initial version of the manuscript. In fact, the research director distributed around 1528 questionnaire among the research team to approach eligible candidates. Some women were either reluctant to join, non-Lebanese, born Lebanese but living abroad, or having difficulties in reading Arabic. Hence, statistical analysis were performed on the total number of complete collected material (N=1113). This issue is clarified in the final revised manuscript (L. 129).

4) I also wonder if excluding the women that did not answer sexual-related questions couldn't be faced as a bias of the study. Couldn't part of the results of these women (the one that they answered) be included in the final analysis?

From the final pool of the sample (N=1113), only 27 women were reluctant to answer questions related on sexual issues, but filled all the other parts of the questionnaires. Their data was included. They represent 2.4% of the total participants.

5) Overall, I don't think it is necessary to explain in details all the questionnaires (e.g., lines 70-77 for MENQOL; lines 80-81 for IPAQ, and so on). Authors should keep it simple and provide references for further details.

Done. We summarized this section.

6) Tables 1 and 2 could be merged.

Done.

7) On all Tables authors should make it very clear the meaning/role of the overwritten letters.

Done.

8) Discussion on lines 232-243 is not necessary as it does not related to the objective of the study.

We aimed to present this information and compare it with other published data, since our study is the first conducted among a large number of women, and no national statistics were available.

9) I think the physical activity is a nice outcome of the study, and this was independently associated with most of the subdomains of quality of life in the participants, which reinforce the importance of this variable to the study conclusions. However, authors should acknowledge the limitations of using a questionnaire to assess physical activity as it might present inaccuracies in comparison with objective measurements (e.g., accelerometers).

Done. This issue was mentioned in the limitation section. 

10) Another limitation is the absence of blood tests to check levels of FSH and estrogen hormones (even if it was done in a sub sample). I would include this in the limitation section but I don't think this limitation reduce the quality of the study.

Blood tests were not included in the methodology of this study. A sentence was added in the limitation section concerning this issue. 

11) It is not clear if there was a sampling strategy (and sample calculation) to generate a sample representative of all Lebanon areas. If not, I would include this as a limitation of the study.

The minimum sample size to be included in this study was calculated initially by following the formula of Tabachnick and Fidell1 (2001) that take into consideration the number of explanatory variables to be include in the model: N = 50 + 8m (m is the number of explanatory variables); Given that m=14, a minimum of 162 women should be recruited in this study. Our sampling strategy was not representative of all Lebanese women since we followed a convenient sampling strategy, however it englobed a large number of participants from different educational, marital and religious backgrounds and of various stages of the menopausal transition.This explanation was added in the final manuscript.

1Tabachnik BG, Fidell LS. Using multivariate statistics. 4th ed. Needham Heights, New York: Harper Collins (2001)

Reviewer #2: I appreciate the effort of the authors to conduct the study for reporting the relationship between physical activity level, menopausal symptoms and the quality of life of Lebanese women. After completling my review, several concerns should be revised to fulfill the quality of the study.

General Comments

Please standardize the objective of the study on the different topics of this article (i.e. abstract, introduction and, discussion).

Done

Please review your reference as more than half of the references listed are very outdated. If able, prefer to keep references not older than 10 years.

Done. We removed 6 old references and added 14 new ones.

In addition, some English mistakes need to be corrected through all the manuscript.

The manuscript was corrected by a colleague, whose native language was English.

Specific Comments

Abstract

In methods, please provide information regarding the classification of menopausal status.

Done. We clarified this classification in the text by following WHO’s division. 

Introduction

Please add bibliographic references in the first sentences of the first paragraph. 

Done. New References were added to the final text in the introduction, such as: 

1. Maric-Bilkan C, Gilbert EL, Ryan MJ. Impact of ovarian function on cardiovascular health in women: focus on hypertension. Int J Womens Health. 2014;6:131–9. 

2. Cauley JA. Estrogen and bone health in men and women. Steroids. 2015 Jul;99(Pt A):11–5. 

3. Bruce D, Rymer J. Symptoms of the menopause. Best Pract Res Clin Obstet Gynaecol. 2009 Feb;23(1):25–32. 

Please add the meaning of “MENQQOL”.

Done. We specified the meaning of Menqol in the final manuscript.

Methods

Please provide information regarding the randomization process of the different regions recruited.

Our sampling strategy was not representative of all Lebanese women since we followed a convenient sampling strategy, however it englobed a large number of participants from different educational, marital and religious backgrounds and of various stages of the menopausal transition. Data emerged from this study can help researches establish future perspectives on national level by applying proper randomization and stratification and adding more parameters. This sentence was added in the limitation section.

Was any participant using hormone replacement therapy?

Women taking hormone replacement therapy were excluded during recruitment.

The authors do not mention the application of a cognitive test. Participants with cognitive impairment may have difficulty with the questionnaires.

Participants suffering from mental disorders such as dementia or cognitive impairment were not invited to participate in the study. Those medical conditions were considered an exclusion criteria, during recruitment.

Results

Please review this session. Some table results are not consistent with the text.

Done. All tables in the result section were corrected and arranged.

Please review the table captions. It is not possible to identify statistical differences among/between groups.

Done. All table captions were corrected an arranged.

Please add to table 4 the results of the physical activity level data analysis.

Done. The results were added. 

Discussion

1° Paragraph: Please discuss more about the impact of anthropometric measurements (e.g. BMI, WC and WHR) on participants of this current study. Or join this part with the beginning of the 4° paragraph.

Done. We joined this part with the beginning of the 4th paragraph.

Please discuss more about the relationship between the results of the activity physical level and the different subdomains of women's quality of life.

Done. We added a new paragraph discussing this relationship (L.815-844).

Please discuss whether the results reach the level of clinical significance.

As presented in Table 5, highly significant statistical results were observed between the physical activity level and the four domains of the MENQOL questionnaire (vasomotor, psychosocial, physical and sexual). Values obtained between the domains and the categorization of physical activity as low, moderate and high presented a good and acceptable level of clinical significance, explained by the fact that women who were physically more active, were experiencing less discomforts in all the tested domains.

---

## [Decision Letter · Decision Letter 1]

19 Feb 2020

PONE-D-19-25613R1

Menopausal symptoms, physical activity level and quality of life of women living in the Mediterranean region.

PLOS ONE

Dear Dr Papazian,

Thank you for submitting your manuscript to PLOS ONE. After careful consideration, we feel that it has merit but does not fully meet PLOS ONE’s publication criteria as it currently stands. Therefore, we invite you to submit a revised version of the manuscript that addresses the points raised during the review process.

We would appreciate receiving your revised manuscript by Apr 04 2020 11:59PM. To enhance the reproducibility of your results, we recommend that if applicable you deposit your laboratory protocols in protocols.io, where a protocol can be assigned its own identifier (DOI) such that it can be cited independently in the future. For instructions see: http://journals.plos.org/plosone/s/submission-guidelines#loc-laboratory-protocols

We look forward to receiving your revised manuscript.

Kind regards,

Anderson Saranz Zago, PhD

Academic Editor

PLOS ONE

Additional Editor Comments (if provided):

Dear Authors

According to the opinion of the reviewers, the manuscript brings an interesting subject, however, it needs to be reviewed on some topics (minor revision). After all these changes, the authors can resubmit the manuscript for a new evaluation.

Sincerely

Anderson Saranz Zago, PhD.

Academic editor

Reviewers' comments:

Reviewer's Responses to Questions

**Comments to the Author**

1. If the authors have adequately addressed your comments raised in a previous round of review and you feel that this manuscript is now acceptable for publication, you may indicate that here to bypass the “Comments to the Author” section, enter your conflict of interest statement in the “Confidential to Editor” section, and submit your "Accept" recommendation.

Reviewer #1: All comments have been addressed

Reviewer #2: All comments have been addressed

2. Is the manuscript technically sound, and do the data support the conclusions?

Reviewer #1: Yes

Reviewer #2: Yes

3. Has the statistical analysis been performed appropriately and rigorously? 

Reviewer #1: Yes

Reviewer #2: Yes

4. Have the authors made all data underlying the findings in their manuscript fully available?

Reviewer #1: Yes

Reviewer #2: Yes

5. Is the manuscript presented in an intelligible fashion and written in standard English?

Reviewer #1: Yes

Reviewer #2: Yes

6. Review Comments to the Author

Reviewer #1: Dear authors,

Thanks for you answers.

I'm fairly satisfied with the answers and modifications.

I only have one remaining comment:

It is not clear what the authors meant by "invalidating" in Line 309.

Thanks

Reviewer #2: In their majority, comments and suggestions made in the first round of the peer review process have been attended. Minor review is still needed to improve the manuscript.

Abstract

Please remove the word “randomly”, since patient recruitment was not randomized.

Please change “postmenopausal in the sexual domain (p<0.001).” to “post-menopausal and menopausal women in the sexual domain (p<0.001)”.

Results

It is still difficult for the reader to understand the statistical differences among / between groups in the tables. I suggest a review them.

I suggest adding a table with the clinical data (e.g. anthropometric measurements, lifestyle, health) according to the menopause status.

7. PLOS authors have the option to publish the peer review history of their article (what does this mean?). If published, this will include your full peer review and any attached files.

Reviewer #1: Yes: Tiago Peçanha

Reviewer #2: No

---

## [Author Response · Author response to Decision Letter 1]

23 Feb 2020

Comments to the Author

 In their majority, comments and suggestions made in the first round of the peer review process have been attended. Minor review is still needed to improve the manuscript.

Abstract

Please remove the word “randomly”, since patient recruitment was not randomized. Done.

Please change “postmenopausal in the sexual domain (p<0.001).” to “post-menopausal and menopausal women in the sexual domain (p<0.001)”. Done.

Results

 It is still difficult for the reader to understand the statistical differences among / between groups in the tables. I suggest a review them. All statistically significant values were put in bold. All tables were reviewed by Dr Nada El Osta, the statistician of this research. Thank you. 

I suggest adding a table with the clinical data (e.g. anthropometric measurements, lifestyle, health) according to the menopause status. A new table was added in the supplementary file (S.2 Table), and was cited in the manuscript. Thank you.

---

## [Editor Report · Decision Letter 2]

3 Mar 2020

Menopausal symptoms, physical activity level and quality of life of women living in the Mediterranean region.

PONE-D-19-25613R2

Dear Dr. Papazian,

We are pleased to inform you that your manuscript has been judged scientifically suitable for publication and will be formally accepted for publication once it complies with all outstanding technical requirements.

With kind regards,

Anderson Saranz Zago, PhD

Academic Editor

PLOS ONE

Additional Editor Comments (optional):

Dear Author

After resubmission, it can be observed that all comments were addressed. So, I am pleased to inform you that your manuscript has been deemed suitable for publication in PLOS ONE. Congratulations!

Sincerely

Anderson Saranz Zago

Academic Editor
---

## [Editor Report · Acceptance letter]

6 Mar 2020

PONE-D-19-25613R2 

Menopausal symptoms, physical activity level and quality of life of women living in the Mediterranean region. 

Dear Dr. Papazian:

I am pleased to inform you that your manuscript has been deemed suitable for publication in PLOS ONE. Congratulations! Your manuscript is now with our production department. 

With kind regards,

on behalf of

Dr. Anderson Saranz Zago 

Academic Editor

PLOS ONE